# Rising Roles of Small Noncoding RNAs in Cotranscriptional Regulation: In Silico Study of miRNA and piRNA Regulatory Network in Humans

**DOI:** 10.3390/genes11050482

**Published:** 2020-04-29

**Authors:** Massimiliano Chetta, Lorena Di Pietro, Nenad Bukvic, Wanda Lattanzi

**Affiliations:** 1U.O.C. Genetica Medica e di Laboratorio, Ospedale Antonio Cardarelli, 80131 Napoli, Italy; 2Dipartimento Scienze della Vita e Sanità Pubblica, Sezione di Biologia Applicata, Università Cattolica del Sacro Cuore, 00168 Rome, Italy; 3UOC Lab. di Genetica Medica, Azienda Ospedaliero Universitaria Consorziale Policlinico di Bari, 70124 Bari, Italy; 4Fondazione Policlinico Universitario A. Gemelli IRCCS, 00168 Rome, Italy

**Keywords:** gene expression regulation, miRNA, piRNA, transcriptional modulation, binding motifs, regulatory pathways, epigenetics, in silico

## Abstract

Gene expression regulation is achieved through an intricate network of molecular interactions, in which trans-acting transcription factors (TFs) and small noncoding RNAs (sncRNAs), including microRNAs (miRNAs) and PIWI-interacting RNAs (piRNAs), play a key role. Recent observations allowed postulating an interplay between TFs and sncRNAs, in that they may possibly share DNA-binding sites. The aim of this study was to analyze the complete subset of miRNA and piRNA sequences stored in the main databases in order to identify the occurrence of conserved motifs and subsequently predict a possible innovative interplay with TFs at a transcriptional level. To this aim, we adopted an original in silico workflow to search motifs and predict interactions within genome-scale regulatory networks. Our results allowed categorizing miRNA and piRNA motifs, with corresponding TFs sharing complementary DNA-binding motifs. The biological interpretation of the gene ontologies of the TFs permitted observing a selective enrichment in developmental pathways, allowing the distribution of miRNA motifs along a topological and chronological frame. In addition, piRNA motifs were categorized for the first time and revealed specific functional implications in somatic tissues. These data might pose experimental hypotheses to be tested in biological models, towards clarifying novel in gene regulatory routes.

## 1. Introduction

Roughly 90% of the genome is transcribed. Protein-coding (structural) genes account for as little as ~2% of the genome, with a much larger portion of the transcribed content being represented by noncoding RNAs (ncRNAs) [1,2]. Since the original characterization of the first ncRNA, a transfer RNA (tRNA) purified from yeast in 1965 [3], the conceptualization of the “RNA world” [4] and the discovery of different types of ncRNAs (including ribosomal RNAs, long and small noncoding RNAs, and circular RNAs) has improved significantly. The understanding of the roles and mechanisms of action of noncoding RNAs (ncRNAs) has been progressively increasing throughout this postgenomic era, completely revolutionizing the idea of “junk DNA” [1,2]. In particular, most small noncoding RNAs (sncRNAs) discovered so far (i.e., microRNAs (miRNAs), PIWI-interacting RNAs (piRNAs), tRNA-derived stress-induced small RNAs (tiRNAs), small-interfering RNAs (siRNAs)) are responsible for a variety of regulatory mechanisms affecting gene expression at multiple levels.

Gene expression is regulated in a tissue-specific fashion with time-related rhythms and stimuli-responsive mechanisms. This is achieved through the fine-tuning orchestrated by a number of molecular interactors and epigenetic regulators that act at different levels. These include known DNA trans-acting elements, such as transcription factors (TFs), mostly binding to promoter elements upstream of genes, and transcriptional activators and repressors, which interact with enhancer and silencer regions, respectively. In addition, different classes of sncRNAs, including miRNAs and piRNAs, perform a dynamic epigenetic regulation and surveillance of genomic integrity and functions, acting on the genome scale and determining a dense and complex regulatory network [5]. 

miRNAs are short, single-stranded noncoding RNAs (20–22 nucleotides in length) that are typically transcribed by the RNA polymerase II. miRNAs are encoded by the major regulatory gene family in eukaryotic cells and operate in multiple signaling pathways [6,7]. In particular, the control of mRNA stability and translation is the best known and characterized miRNA function. According to the consolidated model, mature miRNAs are found and act in the cytoplasm, where they recognize and bind complementary cis-regulatory elements usually located in the 3′UTR of the target mRNAs, through a specific short (6–7 nucleotides) sequence known as ’seed sequence’ in their 5’ end. This interaction determines the degradation of the mRNA or the repression of its translation [5]. miRNAs are commonly categorized in family groups based on their sequence conservation, with specific regard to the seed sequence. Therefore, the miRNA family members considered to date tend to share similar targets and are usually involved in interconnected pathways [8]. 

Increasing evidence points towards the idea that mature miRNAs are also enriched in the nucleus, suggesting noncanonical roles, i.e., in transcriptional silencing or activation and alternative splicing [9,10,11]. The conditions through which miRNAs elicit their nuclear function have not been fully clarified, and the mechanisms that regulate miRNAs’ translocation into the nucleus are also extensively debated [11]. It has been reported that miRNAs can induce gene expression by directly binding to the promoter region, as TFs, through sequence complementarity [9,12]. 

piRNAs are 24–32-nucleotide-long noncoding RNAs, named after their association with the argonaute subfamily PIWI proteins, originally identified in mutants affected by asymmetric division of stem cells in the Drosophila germline [13]. piRNAs are the largest subgroups of sncRNAs recognized in animal cells, accounting for over 30,000 members [14]; this number is increasing rapidly along with the discovery of different piRNA isoforms [15]. They are known to act specifically in germline cells to preserve the genome integrity from the deleterious effects of transposable elements, through both transcriptional and post-transcriptional repression mechanisms (i.e., suppressing transposon activity, protecting the telomere, and regulating RNA silencing and epigenetic control by establishment of a repressive chromatin state) [16]. piRNAs have more recently been found also in somatic cells [17], where their expression is regulated in a tissue-specific manner, although the impact of this discovery is not yet completely understood [18]. 

The defective or aberrant expression of sncRNAs has been associated with several human diseases, and the interest in their biological roles has increased concurrently [18,19].

Advances in methods that analyze RNA populations have allowed a rapid quantitative and qualitative characterization of small RNAs at the cellular and/or tissue level. As a result, a huge quantity of “omics” data are currently available, stored in specialized databases and representing a valuable source of information which could potentially provide surprising and useful details on genome architecture and regulatory frameworks [20]. Precious hints indeed derive from the analysis of sncRNA sequences that could reveal the presence of conserved domains plausibly enabling additional RNA–RNA or RNA–DNA interactions, gathering them in the context of specific cellular functions on a genome-wide scale.

In this scenario, the aim of this study was to analyze the complete subset the whole sequences of miRNAs and piRNAs stored in the main databases in order to identify the occurrence of conserved motifs and subsequently predict possible specific interplay with TFs within genome-scale regulatory networks. We propose a direct interaction between the sncRNAs (either miRNAs or piRNAs) with conserved common motifs (transcription factor binding site (TFBS)) on DNA sequences. This original analytical workflow allowed identifying the occurrence of conserved motifs in both miRNAs and piRNAs and categorizing these sncRNAs based on TFBS domains. The data produced by this in silico pipeline point towards the hypothesis that miRNAs and piRNAs share DNA-binding motifs with TFs. We then propose that these sncRNAs may be categorized based on TFBS. Interestingly, the functional annotation of putative target genes allowed evidencing that these binding motifs are specifically enriched in biological networks involved in embryonic development. Indeed, the analysis allowed pointing out that different miRNA and piRNA classes, sorted by TFBS, have differential implication in biological pathways, as they are able to regulate multiple target genes sharing the same conserved TFBS. Our in silico analysis provides original predictive computational data, paving the way to further biological studies that may test novel nuclear roles of miRNAs and piRNAs in gene expression regulation.

## 2. Materials and Methods 

The analysis pipeline was performed using different bioinformatics tools available online in the MEME-suite collection of motif-bases sequence analysis tools (http://meme-suite.org/). The experimental pipeline consisted of three main steps, as schematized in Figure 1.

The entire subgroup of *Homo sapiens* mature miRNAs (1881 mature miRNA sequences from http://www.mirbase.org/ftp.shtml; only unique sequences were considered) and piRNAs (32,826 piRNA sequences from http://regulatoryrna.org/database/piRNA/download.html) were analyzed separately.

In the first step, all reliable conserved motifs were categorized by searching in miRNA and piRNA sequences using the Discriminative Regular Expression Motif Elicitation (DREME) tool from MEME-suite (http://meme-suite.org/tools/dreme) [21]. The DREME tool allows finding relatively short motifs (up to eight positions) using sets of sequences (in our case, miRNAs and piRNAs, as reported in * fast sequences) as input. The program does not need a control set since it shuffles the primary set to provide it. Moreover, it exploits Fisher’s exact test to determine significance of each motif found in the positive set using a significance threshold. The motifs identified by this approach were stopped when the next motif’s *E*-value threshold exceeded 0.05 (default threshold) [21]. The identified motifs were mapped within the miRNA sequences in order to observe their localization and possible relation with the seed sequences, according to the MicroRNA Target Prediction Database (miRDB; http://www.mirdb.org).

In the second step, all the obtained motifs were used as query for Tomtom, a motif comparison tool within the MEME-suite (http://meme-suite.org/tools/tomtom), which compares the newly identified motifs against a database of known motifs (i.e., JASPAR). JASPAR CORE is a database that contains a curated and nonredundant set of open access data collections of experimentally discovered and proven TF binding sites [22,23]. Tomtom ranked the motifs in the database and produced an alignment for each significant match, searching one or more query motifs against one or more databases of target motifs (and their reverse complements when applicable). The report for each query was a list of target motifs, ranked by *p*-value in the order that the queries appear in the input file. The *E*-value and the q-value for each match were also reported. The q-value is the minimal false discovery rate at which the observed similarity would be considered significant. Tomtom estimated q-values from all the match *p*-values using the Benjamini and Hochberg method. By default, significance was measured by q-value of the match [22].

For all motif queries, a list of transcription factors that contained the common conserved domain was obtained (all the motifs with corresponding TF lists are reported in Table 1 and Table 2).

Finally, in the third step, all lists of putative miRNAs and piRNAs that contained motif-related TFs were individually loaded in GeneMANIA (https://genemania.org/). This is a flexible, user-friendly web interface for generating hypotheses about function, through analyzing gene lists and prioritizing them based on literature-proved biological functions. GeneMANIA allowed clustering functionally related TFs, using available genomic and proteomic data (protein–protein, protein–DNA interactions, signaling pathways, protein domains, and phenotypic screening profiles) introducing weights that indicate the predictive value of each selected dataset for the query [24]. 

The lists of annotated functions were then used for the biological interpretation to achieve a functional hypothesis. In particular, using a Benjamini–Hochberg false discovery ratio (FDR) multiple testing correction (also known as the ‘q-value’) associated with each ‘function’, we selected for the analysis only GO terms with *FDR* ≤ 0.05, in order to reduce redundancies.

## 3. Results

### 3.1. Conserved Motif Annotation in miRNAs and piRNAs

The DREAM tool (step 1 of our workflow, see Figure 1) allowed identifying conserved motifs in both miRNAs and piRNAs. These motifs were predicted to bind specific DNA sequences on the basis of common domains with TFs, suggesting a putative involvement of sncRNAs in nuclear gene regulation. 

In particular, we identified conserved motifs in 66.7% of mature miRNAs and in about 94% of the piRNAs analyzed. Particularly, 14 and 39 conserved motifs were identified in miRNAs and in piRNAs, respectively (Table 1 and Table 2). The analysis was performed considering complementary sequences as well, in order to identify motifs able to bind on both directions and on either DNA strand.

When mapping the precise location of the motifs within the miRNA sequences, we found that these do not necessarily correspond to the seed sequence positions. In some cases, the sequence motif is localized at the 5’ end of the tested miRNA, where it only partially overlaps with the seed sequence. In all other cases, the motifs were mapped in variable regions of the mature miRNA sequences. 

Of the 14 conserved motifs identified in miRNAs, four motifs (RAAAGWAA, CYUUCUG, UGUGAY, and GGAMAG) were present in more than 100 miRNAs when also considering complementary sequences. The least represented motif is AUUACUUU, which can be recognized in as few as 26 miRNAs (entire range: from 26 to 269 miRNAs; see Table 1). 

The number of piRNAs containing one or more of the 39 putative DNA-binding motif ranges from 23 for the ATTGCACG motif to 21,965 for the CAYCW domain (Table 2). It is noteworthy that, in some piRNA sequences, more than one conserved motif has been identified, hence the extent of DNA-binding partners and corresponding target genes would be even greater. 

The Tomtom tool (step 2 of our workflow, see Figure 1) allowed the annotation of the target TFs containing the DNA-binding motifs found in the tested miRNAs (Table 1) and piRNAs (Table 2). Our data showed that while the number of putative transcription factors sharing either of the conserved DNA-binding motifs identified in miRNAs is relevant (Table 1), some of the conserved motifs (namely ACTCGYG and GGTTCCGA) identified in piRNAs could be associated to just one gene (Table 2).

### 3.2. Computational Pathway Analysis of TFs

Through the GeneMANIA tool (step 3 of our workflow, see Figure 1) we were able to predict the functional implication of the putative TFs sharing the motifs identified in miRNAs and piRNAs (listed in Table 1 and Table 2). This allowed speculating on their functional implication and networking in biological processes. The entire lists of TF-related ‘functions’ computed through the GeneMANIA software are provided in Appendix A (miRNA and piRNA motif-associated functions). In order to rationalize the wide range of GO terms associated as ‘functions’ to each TF list, we performed a further additional clustering of similar functional categories by grouping biologically related functions shared by TF lists related to both miRNA and piRNA motifs. Overall, we categorized 644 functions with corresponding GO terms into 21 functional clusters, listed in the Appendix A. Specifically, 12 of 14 miRNA motifs were linked by GeneMANIA to annotated functions, and only seven of these domains could be positively associated within 16 functional clusters, as schematized in Figure 2 (see also Appendix A). Also, 35 of 39 piRNA motifs were associated with biological functions; of these, 29 domains could be arranged in 21 clusters, with five new clusters in addition to those identified for miRNA domains (namely ‘Apoptosis and cellular response to stress stimuli’, ‘Endocrine signaling’, ‘Metabolic processes’, ‘Regulation of circadian rhythm’, and ‘RNA-mediated gene silencing’) (Figure 3, Appendix A).

As expected, nonspecific “DNA-binding” activities, inherently associated to all TFs, accounted for 88 out of the 644 (14%) annotated functions (grouped in the ‘DNA interaction/gene expression regulation’ functional cluster, Appendix A). In particular, these were annotated in all but one (namely GCUUCCHU) of the miRNA motifs and in all but four of the piRNA motifs (namely HATCCTA, TAAGGGTA, GAACGGGY, KGGCTTAA). Furthermore, 118 out of the 644 (18%) GO annotations found for most miRNA and piRNA motifs represented general biological processes and pleiotropic signal transduction pathways that cannot be clearly categorized into univocal biological functions (grouped in the ‘General cellular processes/pleiotropic signal transduction pathways’ functional cluster, Appendix A). Due to the nonspecificity of these annotations, the above specified clusters were not considered further in the biological interpretation. 

Interestingly, after excluding nonspecific functions, as many as 293 functions out of the 644 (45%) GO annotations were associated with specific developmental stages, from early embryo formation to detailed organogenetic paths (Appendix A). We have further described these functional categories and the related sncRNA motifs, in the attempt to provide an experimental hypothesis to be tested in wet lab functional studies (Figure 2 and Figure 3).

In particular, the analysis of the motifs contained in miRNAs (Appendix A) allowed hypothesizing a possible implication of selected motifs in the regulation of specific developmental stages, according to a specific timeframe (Figure 2). In particular, for example, the miRNA sequence motifs RAAAGWAA, UACUUWUG, YGGUUUUU, and AUUACUUU apparently represent putative DNA-binding domains shared not only by TFs involved in the regulation of early developmental stages and morphogenetic events (‘Embryonic development, early stages up to gastrulation’ functional cluster), but also with those involved later events, such as neural, urogenital and skeletal system development or miscellaneous developmental processes (Figure 2). On the contrary, the late stages of embryonic development (‘Embryonic development, late stages, tissue patterning’) seemed to be specifically regulated by the RAAAGWAA motif (Figure 2). Hemopoiesis and endocrine and respiratory system development functions also appeared enriched among the GO terms for the RAAAGWAA motif, while UACUUWUG, ACCAACC, and AUUACUUU domains were found to be involved in the regulation of digestive system development and RAAAGWAA, UACUUWUG, and YGGUUUUU motifs were related to cardiovascular system and muscle development functions (Figure 2). In addition, our in silico analysis showed that RAAAGWAA, GCUUCCHU, GGAMAG, and AUUACUUU binding motifs could affect the expression of genes associated with immune system development and function (Figure 2).

The functional cluster ‘Stem cell homeostasis and differentiation’ (Appendix A) was not considered further in the biological interpretation of miRNA motifs, since this function cannot be associated with a specific timeframe but rather is part of all stages of prenatal and postnatal tissue development and homeostasis. 

Moreover, our analytical workflow also provided evidence of a spatial and temporal distribution for TFs sharing DNA-binding motifs with piRNAs, revealing an intricate network of multiple connections with redundant functions (Figure 3). As already mentioned, the computational analysis of piRNA motifs displayed the enrichment of additional functional clusters, beyond that associated with embryo development, and the regulation of these functions could be thus specifically connected with this class of sncRNAs. Moreover, the gene regulation connected to specific biological processes seemed to be very specific for some types of sequence domains compared with those identified in miRNAs. In further detail, each functional cluster displayed in Figure 3 seemed to be regulated by a characteristic set of DNA-binding domains: if on the one hand it was possible to identify functional clusters regulated by over six motifs (namely ‘Apoptosis and cellular response to stress stimuli’, ‘Immune system development and function’, and ‘Metabolic processes’ functional clusters), other groups appeared to be peculiarly associated with only a single motif (AAAVTGC with hemopoiesis, ABGTTTA with respiratory system development, GCAGAYAC with RNA-mediated gene silencing) (Figure 3). Interestingly, the implication of this piRNA motif in functions related to RNA-interfering mechanisms could directly support the interplay with the annotated TFs and sncRNAs in gene regulation. In addition, for example, some domains were found to be involved in the regulation of circadian rhythm (ACTCGYG, CACGK, ACCWY, ATGAACTC, and GTACGWCA), while others were involved in the regulation of in stem cell homeostasis and differentiation (GCAGAYAC, CCAAAK, CKGCTAAA, and ABGTTTA) (Figure 3).

## 4. Discussion and Conclusions

The complete understanding of the complex miRNA-mediated regulatory network in cells and organisms has not been achieved yet. The selection of molecular interacting targets by miRNA has been long considered to be primarily dictated by sequences at their 5′ end (nucleotides 2 to 7, known as “seed” sequence). Nonetheless, distinct studies have suggested that miRNAs contain additional sequence elements that control their posttranscriptional behavior, including their subcellular localization.

Indeed, it is currently confirmed that mature miRNAs reside in the nucleus, where they participate at several levels of gene expression regulation [9,10,11]. Different pieces of evidence pointed out that RNA-induced silencing complex (RISC) protein complexes also exist in the nucleus, where they actively contribute to the nuclear import of miRNAs (Figure 4) [25,26,27].

The expression enrichment of different miRNA sets in the nucleus seems to vary based on cell type, function, and activity status, or in response to environmental stimuli [12,28]. Numerous efforts have been devoted to the identification of sequence regions within miRNAs able to affect and direct their nuclear import. Hwang and co-workers reported that a hexanucleotide element (AGUGUU) at the 3’ end may affect the subcellular localization of mature miRNAs. This sequence motif apparently acts as a nuclear localization signal enabling the nuclear import of mature miRNA from the cytoplasm [29]. Interestingly this motif resembles the reverse sequence of the UGUGAY motif identified in this study. Also, another study found that two additional sequence motifs were found in nuclear-localized miRNAs expressed by endothelial and muscle cells upon hypoxic conditions [28]. Moreover, another study showed that most of the nucleus-enriched miRNAs share a common sequence motif with homology to the consensus MYC-associated zinc finger protein (MAZ) transcription factor binding element [30].

The noncanonical nuclear role of miRNAs in the regulation of gene expression at the transcriptional level is yet to be fully clarified. Alternative mechanisms have been proposed to date that are not necessarily mutually exclusive but rather suggest that miRNAs may intervene at several levels in the gene expression regulatory network occurring in the nucleus. Most of the described mechanisms members of the nuclear subfamily of argonaute (Ago) proteins, key components of RISC complexes, as key mediators. Such nuclear miRISC complexes may bind long noncoding RNAs (lncRNAs) by sequence complementarity and modulate their function [11,12]. The lncRNA class of ncRNA includes epigenetic mediators acting in the nucleus (including ‘promoter-associated’ and ‘enhancer-associated’ RNAs and ‘gene body-associated’ RNAs). These are in turn able to influence chromatin organization, acting as structural scaffolds of nuclear domains, and to mediate transcriptional/cotranscriptional regulation [31]. Other experimental studies suggested miRNAs to be involved with the ribogenesis process occurring in the nucleolus, while others described their participation in the regulation of alternative splicing (see [11] for a review). 

Converging evidence also showed that miRNAs may modify (either activate or suppress) gene transcription by interacting with chromatin, besides acting at the post-transcriptional level in the cytoplasm [9,32]. Mature miRNAs in the nucleus may indeed directly bind double-stranded DNA within specific target sequences [12,33]. Specifically, it has been reported that miRNAs can form triple-helical structures with specific regions of DNA through either Hoogsteen or reverse Hoogsteen pairings [33]. Nonetheless, the likelihood of their effective occurrence of such pairing modalities in physiological conditions is still widely debated [34,35]. It is instead more likely that miRNAs regulate gene transcription by binding to promoter sequences in an Ago-dependent manner, as demonstrated in a number of studies [32,36,37,38]. Ago proteins are known to act in the nucleus, despite their structure not including a known DNA-binding domain; therefore, their interactions with chromatin and chromosomes might be mediated by miRNAs. In particular, the nuclear Ago1 protein directly interacts with RNA polymerase II (RNAPII) and is preferentially enriched in promoters of transcriptionally active genes [39]. The Ago1–RNAPII interaction decreases if miRNAs are depleted, hence suggesting that Ago1–chromosomal interaction is mediated by miRNAs [39]. 

Alternatively, miRNAs may recognize complementary sequences on nascent RNAs, in a cotranscriptional mechanism, forming double-stranded RNAs that determine the recruitment of protein complexes able to modify chromatin accessibility and thus RNAs levels [11,12]. 

Finally, a third model has been also proposed, according to which miRNA–Ago complexes directly target one of the DNA strands when the target promoter region is in an open configuration during the transcription initiation process [12].

Although numerous studies have indicated that the seed sequence can also mediate the recognition of miRNAs’ nuclear targets, a recent study supports a model in which a miRNA can form a hybrid with promoter region to modulate transcription through its nonseed region [40]. In this scenario, the interplay between miRNAs and TFs has been emerging as a key mechanism within the complex network of transcriptional regulatory networks occurring in the nucleus. Such interactions are believed to rely on the presence of conserved regulatory motifs and are needed to finely tune developmental programs in multicellular organisms [41]. Hence, with the aim of making our results as extensive as possible, we decided to include in our in silico analysis all the sequence motifs independently by the overlapping with the seed sequence. Since the mechanisms of miRNA-mediated gene regulation have not been completely clarified and new models continue to be identified in different experimental conditions, our approach aims to avoid losing some important data.

A similar interplay with TFs has not been explored in piRNAs to date, even though increasing roles for this class of sncRNAs, further preserving genome integrity in germline cells, have been recently recognized. The mechanisms at the base of piRNA biogenesis and function have become increasingly clear (Figure 5), and growing evidence suggests that specific piRNA expression patterns can be recognized in pathological conditions, including cancers [42,43].

Understanding the crosstalk between sncRNAs and TFs at a cotranscriptional level could provide new clues towards the involvement of miRNAs and piRNAs in the control of specific events of gene expression regulation in the nucleus during development in humans.

The computational workflow exploited in this study allowed posing an experimental hypothesis according to which conserved and recurrent sequence motifs found in miRNAs and piRNAs, complementary to transcription factor binding sites (TFBS), could influence the bond and activity of TFs on the same target genes. Our results enabled categorizing different classes of motifs, associated to TFs that have known biological roles, hence predicting the possible biological consequence of the putative miRNA nuclear localization and function. 

The biological interpretation of the enriched functional terms in TFs indeed allowed categorizing miRNA motifs according to their involvement in key steps of the human developmental path, suggesting that different miRNA profiles exist in different developmental stages and vary in their nuclear expression across different tissue types. The direct competition/collaboration with TFs that our data suggest might provide a finer regulatory control and could explain the prompt canalization of genetic programs to maintain and stabilize the phenotypic reproducibility of embryogenesis. This type of interaction could increase the speed and efficiency of response of embryonic cells exposed to continuous differentiation stimuli. Indeed, the established mechanisms of miRNA-mediated expression regulation, based on their biding on the 3’-UTR of target mRNAs in the cytoplasm, inevitably occurs at the post-transcriptional level, while the 5’ end DNA-binding event proposed in this model occurs during or right before transcription [10,11,12]. This can therefore cooperate with several mechanisms to accomplish the finely tuned regulatory network especially needed during early developmental stages.

The results obtained in the in silico analysis of piRNA motifs yielded original data providing a model for additional roles for piRNAs in somatic cells, to be further explored in the wet lab. Indeed, piRNA motifs were also found to be associated to TFs with reproducible functions exerted in somatic tissues, including tissue-specific metabolic pathways. It has been shown that the PIWI–piRNA complex binds its genomic target in euchromatin through a nascent transcript and, in heterochromatin, predominantly through a direct piRNA–DNA interaction [44]. Although the information on piRNA roles in somatic tissues is still limited, these data may contribute to postulating new functional roles, regulatory functions, and towards their translation into the identification of new markers of biological processes and/or diseases.

These data may also suggest a feasible way to categorize functional piRNA subclasses distributed in different tissues, on the basis of the presence of conserved motifs, that could reflect their roles in shared regulatory networks and/or developmental timeframes. Each of these families could intervene in crucial parts of the epigenetic control, maintaining genomic integrity, repressing the mobilization of transposable elements, and regulating the expression of downstream target genes via transcriptional or post-transcriptional mechanisms as already reported by two independent research teams in studies of model organisms [45,46]. These groups independently observed that parental responses to the environment are passed to offspring by small RNAs, suggesting that even environment-related behavioral traits can be passed down through generations by transgenerational epigenetic inheritance (TEI), even though the underlying mechanisms are unclear [45,46].

Our observations, though still preliminary, could propose novel testing hypotheses to be investigated in a biological system, towards the clarification of novel aspects of sncRNA-based epigenetic regulation of cellular functions at the organism level.

Further in vitro analyses will be necessary to support at the functional level the evidence derived from this in silico approach. In particular, in-depth in vitro studies at the genome-wide level will be needed to delve into the subcellular location of each class of motif-grouped sncRNAs and to clarify their involvement in the predicted biological pathways.

The extended knowledge of these novel sncRNA mechanisms of action has a deep translational relevance, considering their extensive application in “theranostics”: a differential expression analysis of these sncRNA sequence motifs could enable identifying tissue- or organ-specific biomarkers of pathway function/dysfunction. On the other hand, targeting RNA metabolism is being exploited as a strategy to recover RNA alterations in a variety of diseases, paving the way to RNA-based therapeutic strategies [19].

## Figures and Tables

**Figure 1 genes-11-00482-f001:**
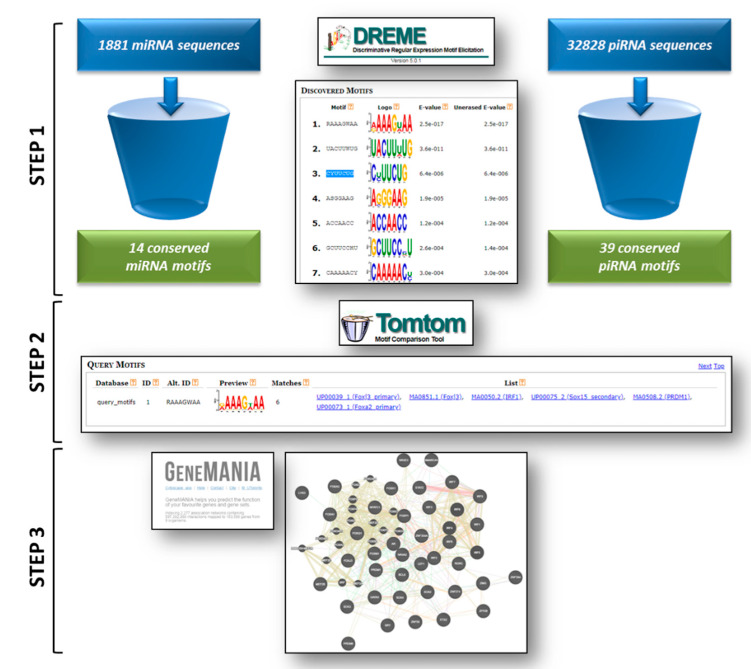
Experimental workflow. A schematic representation of the pipeline integrating the different analytical software is provided. The details of the analytical steps are described in the text (Section 2).

**Figure 2 genes-11-00482-f002:**
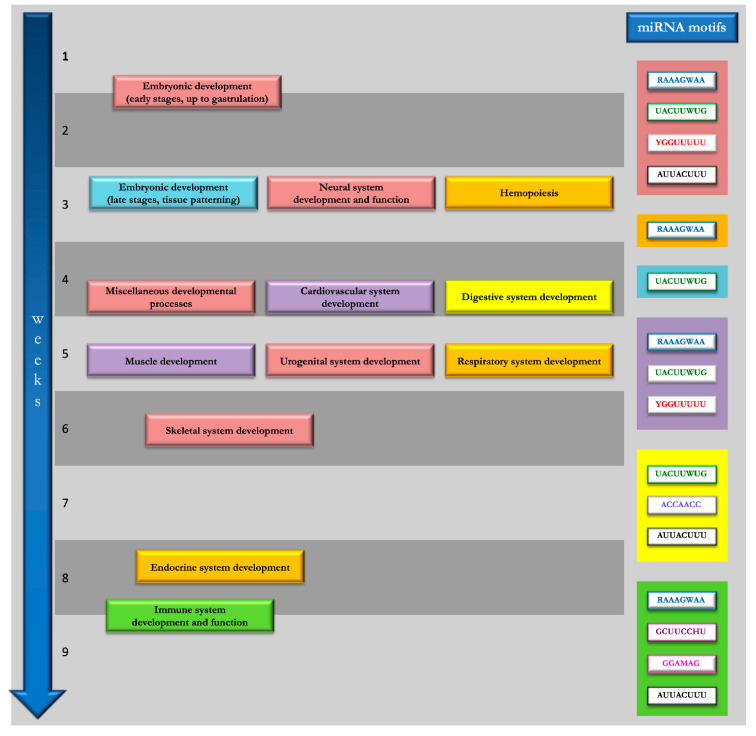
Schematic representation of spatial and temporal distribution of putative DNA-binding motifs identified in microRNAs (miRNAs). The functional clusters identified with the computational pathway analysis of transcription factors (TFs) sharing the motifs identified in miRNAs are depicted as colored boxes distributed along the timeline (arrow on the left side) of human embryo development. The colored boxes (right side; same color ID scale) group the motifs identified in miRNAs for which developmental functions were enriched, according to GeneMANIA functional interpretation (see text for details).

**Figure 3 genes-11-00482-f003:**
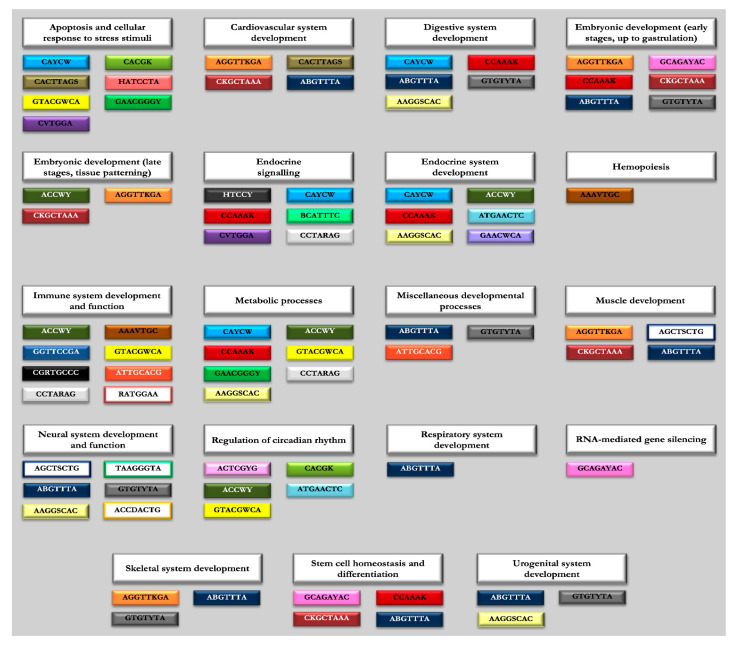
Schematic representation of the enriched functions for PIWI-interacting RNA (piRNA) motifs. The scheme provides an overview of the motifs associated with each functional cluster derived from the computational pathway analysis (based on GeneMANIA tool) of transcription factors (TFs) sharing the same domains with piRNAs (see text for details). A specific colored box is assigned to each piRNA motif.

**Figure 4 genes-11-00482-f004:**
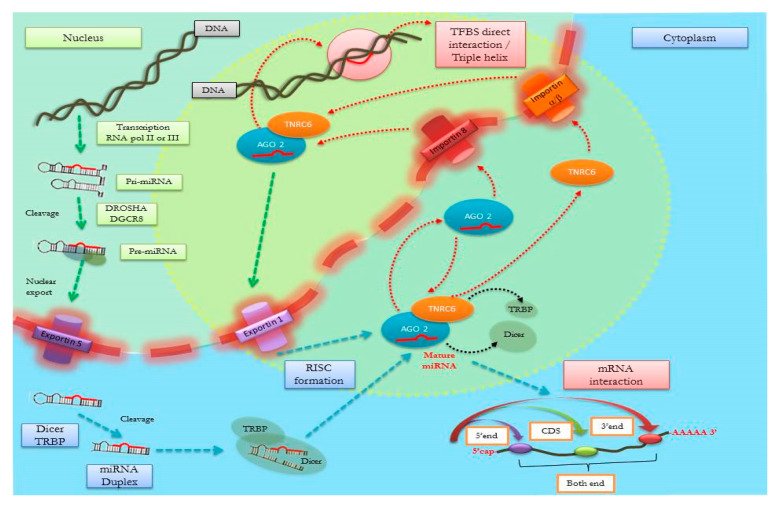
Several steps of microRNA (miRNA) biogenesis and nucleus–cytoplasm transport. Mature miRNAs derive from longer double-stranded primary transcripts (pri-miRNA), which are recognized and processed in the nucleus by the Drosha protein/DGCR8 microprocessor complex subunit (DGCR8) complex into shorter precursors folded in a hairpin loop structure (pre-miRNA). Pre-miRNAs are then exported to the cytoplasm (through Exportin 5) where they are first cleaved by Dicer and later processed by RNA-induced silencing complex (RISC) to form the mature miRNAs. Transactivation response element RNA-binding protein (TRBP) intervenes in the stabilization of Dicer. RISC, which includes Protein argonaute-2 (Ago2), also participates in the identification of miRNAs’ targets. The integrative miRNA network highlights, in the yellow circle, the import of Ago2 with mature miRNA into the nucleus via Importin 8 and trinucleotide repeat-containing gene 6A protein (TNRC6), another component of RISC complex, via Importin β. Nuclear RISC is again assembled to elicit pleiotropic effects by regulating multiple pathways with a direct interaction on DNA transcription factor binding sites (TFBSs) and possible formation of triple-helix structures.

**Figure 5 genes-11-00482-f005:**
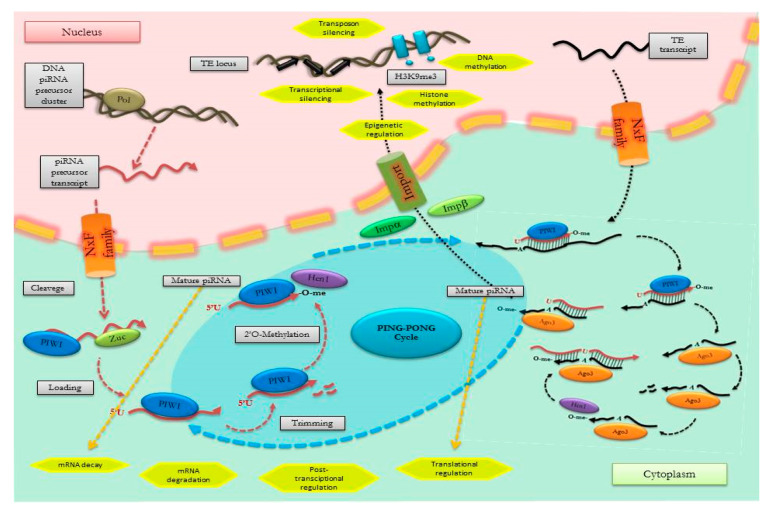
Different stages of PIWI-interacting RNA (piRNA) biogenesis and function. Mature piRNAs are derived from precursor RNAs following a post-transcriptional processing through two alternative mechanisms. The primary maturation pathway involves cleavage of long, single-stranded piRNA clusters and the binding with PIWI proteins in the cytoplasm. The second mechanism is a self-amplifying loop (termed “ping-pong” cycle), in which an antisense piRNA binds PIWI proteins and triggers production of a sense piRNA that binds to Protein argonaute-3 (Ago 3). Nuclear PIWI/piRNA complexes regulate gene and transposon expression by epigenetic modifications. Once the piRNAs are loaded onto PIWI, the activity and/or expression of DNA methyltransferases (Hen 1) is increased, promoting methylation of promoter regions, preventing transcription factor binding, and interacting with histone methyltransferase. Cytoplasmic mature piRNA promotes mRNA decay by interacting with deadenylation complex, inhibits translation by directly binding with translation factors, and modulates cellular signaling by directly regulating the post-translational modifications.

**Table 1 genes-11-00482-t001:** MicroRNA (miRNA) conserved motifs.

Motif Sequence	miRNA-Contained Sequence Motif	miRNA-Contained Complement Sequence Motif	Transcription Factors
RAAAGWAA	100	8	AR, BCL6, ETS2, FOXA1, FOXA3, FOXJ3, FOXK1, FOXM1, FOXP1, FOXQ1, GATA3, IRF1, IRF2, IRF3, IRF7, IRF8, IRF9, LEF1, LHX3, MEF2B, NFATC1, NR2E3, NR4A2, PRDM1, PRDM6, RORG, SMARCA1, SOX2, SOX3, SOX4, SRY, STAT2, ZFP28, ZIM3, ZNF274, ZNF354A, ZNF394, ZNF85
UACUUWUG	59	2	CDX1, FOXA1, FOXA2, FOXA3, FOXC1, FOXM1, LEF1, LHX3, MEF2A, MEF2B, MEF2C, MEF2D, NR2E3, POU5F1, ZNF708
ACCAACC	33	1	ARI5B, FOXI1, GLI3, NFYA, RUNX2, SALL4, Z324A, ZIC1, ZN449
CYUUCUG	106	33	ATOH1, BCL6, EHF, ERG, ETS1, ETV4, NDF1, NDF2, NR1D1, OLIG2, OSR2, PRGR, SMAD4, STAT2, THA11, ZIC3, ZIM3, ZN143, ZN436, ZN528, ZN547, ZN768, ZNF76, ZSC31
ASGGAAG	60	10	E2F6, ELF1, ELF2, ELK1, ELK4, ETS2, ETV1, FEV, GABPA, IRF3, NFKB1, OLIG2, PAX6, T, ZNF257, ZNF341, ZNF436, ZNF528, ZSCAN31
GCUUCCHU	41	4	ELF3, ERG, ETS2, FEV, FEZF1, FLI1, FOXK1, NHLH1, NFKB1, NFKB2, OLIG2, SMARCA1, SMARCA5, SOX10, ZNF341, ZNF394, ZNF436, ZNF502, ZNF528, ZNF582, ZNF85
CAAAAACY	35	2	HOXA10, NFATC3, RUNX2, ZIM3, ZNF384
UUACBGU	40	4	ELK1, ETS1, ETV4, OVOL1, RFX3, ZIM3
YGGUUUUU	37	3	AIRE, FOXO4, FOXQ1, HXA10, HXA13, MEF2C, NR2E3, RUNX1, RUNX2, RUNX3, TWST1, ZN384
UGUGAY	179	90	ERR1, RXRB, GFI1, GFI1B, MITF, TFEB, JUNB, TFE3, ESR1, RARG, FOXI1, USF1, ZIC3
KAGGUUG	74	21	DUX4, HIC1, ZIM3, ZN136, ZN768
GGKUGGGG	48	9	E2F4, E2F6, EGR1, EGR2, ESR2, GLI3, KLF1, KLF12, KLF15, KLF3, KLF4, KLF5, KLF6, KLF9, MAZ, MXI1, PATZ1, PRDM14, RXRA, SALL4, SP1, SP2, SP3, SP4, SREBF1, SREBF2, TAL1, TBX3, VEZF1, WT1, ZBTB17, ZIC1, ZNF281, ZNF449, ZNF467
GGAMAG	153	78	BCL6, E2F1, E2F4, E2F6, E2F7, ERG, ETS1, ETV4, GATA1, IRF3, MEIS2, MYOD1, MYOG, NFATC1, NFATC2, NFATC3, NFATC4, NFKB1, NFKB2, NR1D1, NR1I3, PBX1, PRDM1, RELB, REST, STAT1, RELA, TFDP1, TGIF1, ZNF257, ZNF274, ZNF335
AUUACUUU	25	1	ALX1, CDX2, DUX4, EVX2, FOXA1, FOXA2, FOXA3, FOXC1, FOXK1, FOXM1, HNF6, IRF1, IRF2, IRF7, IRF8, LHX2, LHX3, NR2E3, PBX2, PRDM1, ZFP28, ZN394, ZNF85

**Table 2 genes-11-00482-t002:** PIWI-interacting RNA (piRNA) conserved motifs.

Motif Sequence	piRNA-Contained Sequence Motif	piRNA-Contained Complement Sequence Motif	Transcription Factors
HTTCY	11887	9673	BCL6, E2F1, E2F3, EHF, ELF1, ELF2, ELF3, ELF5, ELK1, ELK4, ERG, ETS1, ETS2, ETV1, ETV2, ETV4, ETV5, FEV, FLI1, GABPA, HSF1, NFATC2, NFATC3, NFKB1, OSR2, PRDM6, SMARCA1, SP4, STAT5A, STAT5B, STAT1, STAT3, STAT4, STAT6, TFDP1, ZFP28, ZNF317, ZNF394, ZNF418, ZNF528, ZNF680
CAYCW	11906	10059	PDX1, CREB1, SNAI1, TBX3, CEBPG, ATF4, SNAI2, REST, ZEB1, NDF1, NDF2
ACTCGYG	345	55	CLOCK
CGTWCCCA	187	9	NFKB1, NFKB2, RFX1, RFX2, RFX3, RBPJ
CACGK	1749	1151	ARNT, ATF3, ATF6A, BHLHE40, BMAL1, CLOCK, EPAS1, HIF1A, MAX, MITF, MTF1, MXI1, MYC, MYCN, TFE3, TFEB, USF1, USF2
GACKCCTC	204	47	BACH2, CRX, FOSB, FOSL1, FOSL2, JUN, JUND, ZNF317
ACCWY	4967	4083	NR2F1, NR2F2, ERR2, ESRRG, GLI3, NR2C1, NR4A1, NR4A2, PPARG, REST, RUNX1, RUNX2, RUNX3, RXRA, RXRG, TBX21, TBX3, ZIC1, ZNF250, ZNF8
AGGTTKGA	162	26	TBX3, PRDM14, ZNF324, ZNF449, TBX21, TEAD1
AAAVTGC	412	215	MAFF, NKX3-1, MAF, IRF1, IRF2, ZNF85, IRF7, PRDM1, BATF3, HNF4A, STAT2, HNF4G, FEZF1, MYB, HIF1A, RORG, ZFP42
GGTTCCGA	56	2	NR2C2
GCAGAYAC	143	43	CLOCK, SMAD2, ZNF708, MAFB, FOXM1, MXI1, ZFP42, ZNF547
AGCTSCTG	250	111	BHLHA15, MYOG, TCF3, MYOD1, MYF6, HTF4, ITF2, ATOH1, ZBTB18, ASCL1, OSR2, PTF1A, NFE2L2, NHLH1, MAF, RFX5, ZNF563, CEBPG, ATF4, LYL1, ZIC3, NEUROD2, MAFG, OLIG2, TFAP4
CACTTAGS	116	30	NKX3-1, ISL1, NKX3-2, DLX3, FOXA2, PRRX2, BACH2, NFE2, NOBOX, BACH1, MYC, FOXA1, NFE2L2, MYCN, ARNT
HATCCTA	290	147	ZNF586, CRX, ZNF324, ZNF274
AGGYAG	1145	843	ZNF335, ZNF490, MYB, RFX5, ZNF770, ETS1, ERG, FLI1, SMAD3, ZNF257, ZNF549, PTF1A, ZNF563, ETV2
CCAAAK	911	649	HNF4A, HNF4G, FOXA1, ISL1
ATGAACTC	70	13	NR1I3, NR1I2, NR2C1, VDR, ATF2, ZNF18, RARA, ZNF549, RXRB, NR6A1, ZNF354A, NR1H4
CKGCTAAA	67	12	T, ZNF322
BCATTTC	217	105	BCL11A, EHF, ELF1, ELF2, ELF3, ELF5, ELK1, ELK4, ERG, ETS1, ETV1, ETV2, ETV4, ETV5, FEZF1, GABPA, MAFF, NFKB1, POU2F1, POU2F2, POU3F1, POU3F2, REL, SPI1, SPIB, STAT5B, STAT1, STAT2, STAT3, RELA, THRA, ZNF354A, ZFP42, ZFP82, ZNF140, ZNF528
TAAGGGTA	48	5	ZNF264, NKX3-2, CRX, ZNF667, DUX4
GTACGWCA	45	4	EPAS1, ARNT, CREB1, ATF1, RORG, HIF1A, CREM, BHE40, ATF3
CGRTGCCC	44	4	HIC1, NFIA, CUX1
GAACGGGY	41	3	HIF1A, ZBT14, ZNF18
CVTGGA	1343	1057	ZNF436, TEAD4, BCL6, RARG, REST, TEAD1, STAT1, STAT5A, STAT3, ZBTB6, STAT4, SNAI1, POU3F2, TP63, ESRRG
GTAGCTAS	55	9	RFX5, BHLHA15
ATCGCTGA	47	6	NFE2L2, MAFK, MAFG, OZF, ZFP42, ZNF335, ZNF528
ABGTTTA	124	50	FOXC1, FOXJ3, FOXA3, FOXK1, FOXO4, FOXQ1, FOXA1, FOXA2, MEF2D, HNF1B, FOXJ2, FOXP2, HNF1A, FOXO3, SRY, MEF2C, MEF2A, FOXP1, MEF2B, FOXO1, FOXM1
CCAMTAAC	60	14	CREM, ETS1, FOXI1, HOXA13, HOXB13, HOXB4, ISL1, RUNX2, VDR
KGGCTTA	219	120	ZNF528, ZNF41, OTX2, ZNF449, ZNF214
GTGTYTA	202	108	CLOCK, FOXA1, FOXA2, FOXJ2, FOXJ3, FOXK1, FOXO3, FOXO4, FOXP1, FOXP2, FOXQ1, MEF2B, NKX3-1, PRDM14
ATTGCACG	23	//	CEBPA, CEBPD, CEBPB, ATF4, CEBPG
CCTARAG	186	99	BCL6, ZNF436, STAT5A
AKGAGGAC	108	46	ZNF816, ZNF586, THRB, ELF5, ZKSCAN1, CRX, ATF2, ZNF263, ETV5, ZIC3, SPI1, RXRB
AAGGSCAC	80	29	ZNF667, ZNF264, NR4A2, ERR1, NR4A1, STF1, RXRG, NR5A2, TFAP2C, ERR2, ZNF214, ESRRG, HNF4G, SOX9, TFAP2A, NR1H3, RXRB, HNF4A
RATGGAA	87	34	ZNF502, SMARCA5, ZFP82, ZNF582, ZNF394, POU2F1, NFATC1, ZNF264, ZNF354A, FOXK1, ZFP28, HOXA1, BATF3, POU3F1, ISL1, ZNF8, ZNF85, TWIST1, STAT2, NFKB1, IRF3, SOX2, NFATC3, ATF4, NFATC2, PRDM6, CEBPG, ETS2
ACTGWTCG	34	5	ZNF691, CUX1
GAACWCA	338	226	NR3C1, VDR, RXRB, PGR, SOX4, ZKSCAN1, FOXP1
CCGTAGCY	29	3	MYOG, RFX1, RFX2, MYCN
ACCDACTG	104	45	ZKSCAN1, MYB, NEUROD1, SNAI1, KLF8, NEUROD2, ATOH1, TFAP4, HTF4, BHLHA15, NR1D1

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
