# Peer review of "Rising Roles of Small Noncoding RNAs in Cotranscriptional Regulation: In Silico Study of miRNA and piRNA Regulatory Network in Humans"

_genes, 2020, doi:10.3390/genes11050482_

Round 1
Reviewer 1 Report
Massimiliano Chetta and co-workers greatly improved the first draft of the manuscript, they added a better description of the analytical workflow to identify conserved motifs in miRNA and piRNAs and potentially predict direct binding to DNA to suggest possibly novel function in gene expression associated with iRNA non-canonical role. Overall the article is interesting, well written and would provide a valuable resource to propose new roles of human miRNA and piRNAs involved in regulation of gene expression in the nucleus. However, in my opinion, the weakness of the study as the authors recognize in this new version is that it is merely bioinformatic, consequently, the results and conclusions are speculative and need to be supported by additional experimentation. In this particular point, they reply "We are not currently in the situation of performing an in vitro validation, which would be extremely hard to be accomplished on the genome-wide scale." And again, I think it is essential to support their finding that they select at least one or two examples of their best candidates and design and experiment accordingly.
Please correct minor typos I think are given by the editing during the whole text.
Author Response
Massimiliano Chetta and co-workers greatly improved the first draft of the manuscript, they added a better description of the analytical workflow to identify conserved motifs in miRNA and piRNAs and potentially predict direct binding to DNA to suggest possibly novel function in gene expression associated with iRNA non-canonical role. Overall the article is interesting, well written and would provide a valuable resource to propose new roles of human miRNA and piRNAs involved in regulation of gene expression in the nucleus. However, in my opinion, the weakness of the study as the authors recognize in this new version is that it is merely bioinformatic, consequently, the results and conclusions are speculative and need to be supported by additional experimentation. In this particular point, they reply "We are not currently in the situation of performing an in vitro validation, which would be extremely hard to be accomplished on the genome-wide scale." And again, I think it is essential to support their finding that they select at least one or two examples of their best candidates and design and experiment accordingly.
We thank the Referee for his/her opinion, and we are glad to hear that he/she has found the manuscript improved. Regarding the last sentence of the comment, we do understand the concern and we profoundly regret to not be able to move further with experimental testing. In order to provide more convincing data to support our hypothesis, we included an extended discussion of additional evidence in the literature (see the widely revised discussion section).
We also wish to stress again that, in order to thoroughly validate our hypothesis experimentally, we would need to perform experiments on the genome-wide scale and in different cellular models. We believe this also in the light of the many diversified possible nuclear mechanisms of action of sncRNA, which depends on cell type and on the intrinsic and extrinsic factors that modify cell homeostasis (we have added a mention to this in the discussion section, lines 296-297, 428-431, of the present revised version of the manuscript).
Please correct minor typos I think are given by the editing during the whole text.
All typos, misspelling and grammar errors have been corrected, accordingly.
Reviewer 2 Report
In their manuscript Chetta et al. apply an in silico strategy to identify possible sncRNA-TF-links at a genomic-wide and tissue-specific scale. The strategy itself is sound and might prove useful to test the nuclear role of sncRNAs in future studies.
I recommend the manuscript for publication in 'Genes'. However, I would rephrase the sentence in line 472 to 'Further functional analyses will be necessary to support the evidence derived from this...'.
Author Response
In their manuscript Chetta et al. apply an in silico strategy to identify possible sncRNA-TF-links at a genomic-wide and tissue-specific scale. The strategy itself is sound and might prove useful to test the nuclear role of sncRNAs in future studies.
I recommend the manuscript for publication in 'Genes'. However, I would rephrase the sentence in line 472 to 'Further functional analyses will be necessary to support the evidence derived from this...'.
We thank the Referee for the approval of the content of the revised manuscript and for the latter suggestion. We have rephrased the indicated sentences accordingly (lines 428-431).
Reviewer 3 Report
Authors have improved the text and readability of the manuscript. However, there are still major issues still unresolved:
1) Major problem of including the whole miR sequence to analysis still persists. Here are direct quotes from citation provided by authors in their cover letter:
"Interestingly, ChIP-qPCR demonstrates that there is Ago2 at the enhancer locus. Furthermore, no transcriptional activation of neighboring genes of miR-24-1 will occur if Ago2 is knocked down [71]. So, it could be argued that enhancer activation induced by miRNAs requires Ago2 to function directly at the locus or carry mature miRNA from cytoplasm to the nucleus."
and
"The basic principle of microRNA target prediction algorithms is the complement of 5′ end of miRNA and target sequence. As the validation of experimental miRNAmRNA interaction, several empirical miRNA seed sequence models have been proposed, such as nucleotides from position 2 to 8 in the 5′ end of the miRNA [72]. There is possibly an analogous seed model to be exploited for nuclear miRNA target prediction, as some study shows that the mutation or deletion of seed sequence disrupt the activity of nuclear miRNAs [71, 73]."
Therefore, to warrant analysis done in this manuscript, authors must either prove action of free, unbound nuclear miR to genomic DNA or show that nuclear miRs are loaded to Ago in non-canonical way and that targeting is not dependent on seed sequence of miR.
Otherwise analysis of nuclear miR targeting must focus on seed sequence.
2) Hoogsteen paring is still considered as the least likely possibility in the field and therefore other two possibilities of nuclear miR action must be taken in to account in the analysis
3) There are several studies (cited in the table of reference provided in cover letter and also published more recently) that show that some miRs are (a) cytoplasmic enriched, (b) some are found either in cytoplasm or nucleus (depending of eg. of physiological situation) and (c) some miRs are preferentially enriched in nucleus. Therefore, if authors would perform their analysis focused on latter two groups, the results would include much less noise and lower FRD, which is high anyways in in silico analysis.
Author Response
Authors have improved the text and readability of the manuscript. However, there are still major issues still unresolved:
1) Major problem of including the whole miR sequence to analysis still persists. Here are direct quotes from citation provided by authors in their cover letter: "Interestingly, ChIP-qPCR demonstrates that there is Ago2 at the enhancer locus. Furthermore, no transcriptional activation of neighboring genes of miR-24-1 will occur if Ago2 is knocked down [71]. So, it could be argued that enhancer activation induced by miRNAs requires Ago2 to function directly at the locus or carry mature miRNA from cytoplasm to the nucleus." And "The basic principle of microRNA target prediction algorithms is the complement of 5′ end of miRNA and target sequence. As the validation of experimental miRNAmRNA interaction, several empirical miRNA seed sequence models have been proposed, such as nucleotides from position 2 to 8 in the 5′ end of the miRNA [72]. There is possibly an analogous seed model to be exploited for nuclear miRNA target prediction, as some study shows that the mutation or deletion of seed sequence disrupt the activity of nuclear miRNAs [71, 73]." Therefore, to warrant analysis done in this manuscript, authors must either prove action of free, unbound nuclear miR to genomic DNA or show that nuclear miRs are loaded to Ago in non-canonical way and that targeting is not dependent on seed sequence of miR. Otherwise analysis of nuclear miR targeting must focus on seed sequence.
The Authors thank the Referee for the insightful observations and suggestions. Therefore, to better clarify these aspects we have implemented the discussion section by providing more details on the possible alternative ways through which miRNAs are described to interact with the DNA in the nucleus, including the possible role of the seed sequence and other alternative domains. Please see the widely revised discussion section and, particularly the discussion of specific scientific papers, suggesting that if “numerous studies have indicated that the seed sequence can mediate also the recognition of miRNAs’ nuclear targets, a recent study support a model in which a miRNA can form hybrid with promoter region to modulate transcription through its non-seed region (Miao, et al. A dual inhibition: MicroRNA-552 suppresses both transcription and translation of cytochrome P450 2E1. Biochim. Biophys. Acta - Gene Regul. Mech. 2016, 1859, 650–662)” (lines 341-343). Moreover we clearly specify that “with the aim of making our results as extensive as possible, we decided to include in our in silico analysis all the sequence motifs independently by the overlapping with the seed sequence. Since the mechanisms of miRNA-mediated gene regulation has not been completely clarified and new models continue to be identified in different experimental condition, our approach aims to avoid losing some important data” (lines 347-351). In addition, more details have been provided in Materials and Methods (lines 135-137) and Results (lines 180-184) sections, regarding the position comparing of seed sequences and the sequence motifs identified in our study.
2) Hoogsteen paring is still considered as the least likely possibility in the field and therefore other two possibilities of nuclear miR action must be taken in to account in the analysis
We thank the Referee for the suggestion. Several miRNA mechanisms of action have been taken into account and thoroughly discussed in the discussion section of the revised manuscript. In particular, the first part of the discussion have been extensively revised in order to describe a more detailed framework of the knowledge regarding this aspect to date (lines 286-340).
3) There are several studies (cited in the table of reference provided in cover letter and also published more recently) that show that some miRs are (a) cytoplasmic enriched, (b) some are found either in cytoplasm or nucleus (depending of eg. of physiological situation) and (c) some miRs are preferentially enriched in nucleus. Therefore, if authors would perform their analysis focused on latter two groups, the results would include much less noise and lower FRD, which is high anyways in in silico analysis.
The Authors thank the Referee for this precious hint. We actually believe that the identification of an univocal and comprehensive set of miRNAs enriched in the nucleus would be hardly selected for this analysis. Indeed, the profile of miRNA and their relative subcellular distribution/enrichment is extremely dynamic and poorly reproducible in different biological systems, as it depends upon the cell type and functional status, plus on a number of intrinsic and extrinsic factors that modify cell homeostasis. This aspect has been only partially elucidated for selected miRNAs to date, showing this extreme heterogeneity of miRNA localization and functions in different experimental models and conditions (e.g. see ref 28-30). We have discussed this in the present version of the manuscript: “the expression enrichment of different miRNA sets in the nucleus seems to vary based on cell type, function, and activity status, or in response to environmental stimuli [12,28]” (lines 296-297). On a similar note, we have also included selected pieces of evidence from the literature suggesting the expression and functions of piRNAs in somatic cells (lines 354-356).
Round 2
Reviewer 3 Report
Authors have now done substantial improvement of this manuscript in aspects of earlier critic.
This manuscript is a resubmission of an earlier submission. The following is a list of the peer review reports and author responses from that submission.
Round 1
Reviewer 1 Report
Massimiliano Chetta and co-workers present an analytical workflow to identify conserved motifs in miRNA and piRNAs and potentially predict direct binding to DNA to suggest possibly novel function in gene expression associated with iRNA non-canonical role. Overall the article is interesting, well written and would provide a valuable resource to propose new roles of human miRNA and piRNAs involved in regulation of gene expression in the nucleus. However, the weakness of the study is that it is merely bioinformatic, does not provide any controls, filters or inductive reasoning. Therefore, the results and conclusions are speculative and need to be supported by additional experimentation. Points that must be addressed before going any further in the publication process:
It is becoming evident that miRNAs also have specific nuclear functions. Among these, the most studied and still debated activity is the miRNA-guided transcriptional control of gene expression. Although available data detail quite precisely the effectors of this activity, the mechanisms by which miRNAs identify their gene targets to control transcription are still a matter of debate. Chetta et al., mentioned in lines 64-76 that miRNAs are expected to shuttle between the cytoplasm and the nucleus. The question then is whether these miRNAs are active in the nucleus. How do they know? For piRNA, it has been more extensively reported that they have both the nucleus and cytoplasm activity. However, piRNA (and miRNAs) functionality requires several proteins, the majority of which localize to non-membranous organelles. Therefore, the importance of showing by other methods i.e., in situ RNA hybridization experiments that piRNA and targets co-localized and interacting to propose a new function in gene expression regulation. I think it is essential to support their finding that they select at least one or two examples of their best candidates and design and experiment accordingly. Materials and methods are poorly described. Just reading the information provided seems that miRNA and piRNAs were analyzed together (lines 145-147). Which I think was not the case. When they talk about DREME, how do they define the length of 8-nt? This information is essential to find motifs and prevent for misleading or bias the analysis. Please describe how you conclude in defining this window. Still with Materials and Methods. Please describe if only human miRNAs were taking into account or everything in the miRNA was used as is described now. If it was not the case, results are not specifically for humans as the article describes. More Materials and Methods. miRbase has confirmed (functional) and predicted miRNA. Please described how the authors use this information for the analysis. Which controls (out of target miRNA/piRNAs) were taken into account to prove null hypothesis. Lines 292 to 295. “The innovative computational workflow reported in this study put in evidence a new direct interaction of miRNAs and piRNAs on conserved DNA sequence motifs…” How this study put in evidence direct interaction??? In my opinion, this was never shown in this study. The analysis performed if any, would just suggest possible sequence complementarity that needs to be confirmed by other multiple methods. Stiff sentences like this need to be avoided when data is not provided.Minor points:
Line 34. Please delete aka, is unnecessary and informal. Line 64-66. Please correct the examples of the non-canonical roles of miRNAs. The ones listed are recognized as cytoplasmic functions.
Reviewer 2 Report
In their manuscript Chetta et al. apply an in silico strategy to identify possible sncRNA-TF-links at a genomic-wide and tissue-specific scale. The strategy itself is sound and might prove useful to test the nuclear role of sncRNAs in future studies.
Minor comments:
Line 48: It’s usually not the 5’UTR TFs are binding to. Maybe better rephrase it to: ‘binding to promoter elements 5’ of genes’ or ‘upstream of genes’. The same accounts for the next sentence. UTR is misleading here.
General: The results in this manuscript are displayed in a very general manner. This means that it would be better to include some exemplary miRNAs/piRNAs to illustrate the potential influence in a better way. Maybe back this up with a few published examples (have they been found during the pipeline used here?), just to strengthen the hypothesis and the overall value.
General: Did the authors account for biased sequence distribution by e.g. overrepresented miRNA families (e.g. let-7 family)? This might lead to distribution biases that affect sequence motif identification. If not, this should be stated in the manuscript or corrected in the data output.
Reviewer 3 Report
First of all, I am a believer in nuclear miRNAs and appreciate that this is a overlooked but important biology of small RNAs. However, this manuscript has major flaws in study design and execution. It is not clear did authors analyze potential seed sequence binding with TF:s or the whole mature miRNA sequence from public databanks. To make any sense for this kind of analysis the potential miRNA/TF binding site -interactions, they must be analyzed from miRNA profiling studies where sample have been separated to nuclear and cytoplasmic fractions.
Another major flaw is that authors have presumed RNA:DNA triplex binding (Hoogsteen paring not likely in physiological conditions), and negleted very likely miRNA binding to nuclear longer non-coding RNAs (promoter associate or enhancer RNAs). There are several published datasets available for such analysis.
For these reasons I think authors conclusions of associated pathways is invalid and could potentially harm future studies in the associated fields.